# YFORMER: U-NET INSPIRED TRANSFORMER ARCHITECTURE FOR FAR HORIZON TIME SERIES FORECASTING

## ABSTRACT

Time series data is ubiquitous in research as well as in a wide variety of industrial applications. Effectively analyzing the available historical data and providing insights into the far future allows us to make effective decisions. Recent research has witnessed the superior performance of transformer-based architectures, especially in the regime of far horizon time series forecasting. However, the current state of the art sparse Transformer architectures fail to couple down- and upsampling procedures to produce outputs in a similar resolution as the input. We propose the Yformer model, based on a novel Y-shaped encoder-decoder architecture that (1) uses direct connection from the downscaled encoder layer to the corresponding upsampled decoder layer in a U-Net inspired architecture, (2) Combines the downscaling/upsampling with sparse attention to capture long-range effects, and (3) stabilizes the encoder-decoder stacks with the addition of an auxiliary reconstruction loss. Extensive experiments have been conducted with relevant baselines on four benchmark datasets, demonstrating an average improvement of 19.82, 18.41 percentage MSE and 13.62, 11.85 percentage MAE in comparison to the current state of the art for the univariate and the multivariate settings respectively.

## 1 INTRODUCTION

In the most simple case, time series forecasting deals with a scalar time-varying signal and aims to predict or forecast its values in the near future; for example, countless applications in finance, healthcare, production automatization, etc. (Carta et al., 2021; Cao et al., 2018; Sagheer & Kotb, 2019) can benefit from an accurate forecasting solution. Often not just a single scalar signal is of interest, but multiple at once, and further time-varying signals are available and even *known for the future.* For example, suppose one aims to forecast the energy consumption of a house, it likely depends on the social time that one seeks to forecast for (such as the next hour or day), and also on features of these time points (such as weekday, daylight, etc.), which are known already for the future. This is also the case in model predictive control (Camacho & Alba, 2013), where one is interested to forecast the expected value realized by some planned action, then this action is also known at the time of forecast. More generally, time series forecasting, nowadays deals with quadruples $(x, y, x', y')$ of known past predictors $x$, known past targets $y$, known future predictors $x'$ and sought future targets $y'$. (Figure 3 in appendix section A provides a simple illustration)

Time series problems can often be addressed by methods developed initially for images, treating them as 1-dimensional images. Especially for time-series classification many typical time series encoder architectures have been adapted from models for images (Wang et al., 2017; Zou et al., 2019). Time series forecasting then is closely related to image outpainting (Van Hoorick, 2019), the task to predict how an image likely extends to the left, right, top or bottom, as well as to the more well-known task of image segmentation, where for each input pixel, an output pixel has to be predicted, whose channels encode pixel-wise classes such as vehicle, road, pedestrian say for road scenes. Time series forecasting combines aspects from both problem settings: information about targets from shifted positions (e.g., the past targets $y$ as in image outpainting) and information about other channels from the same positions (e.g., the future predictors $x'$ as in image segmentation). One of the most successful, principled architectures for the image segmentation task are U-Nets introduced in Ronneberger et al. (2015), an architecture that successively downsamples / coarsens

its inputs and then upsamples / refines the latent representation with deconvolutions also using the latent representations of the same detail level, tightly coupling down- and upsampling procedures and thus yielding latent features on the same resolution as the inputs.

Following the great success in Natural Language Processing (NLP) applications, attention-based, esp. transformer-based architectures (Vaswani et al., 2017) that model pairwise interactions between sequence elements have been recently adapted for time series forecasting. One of the significant challenges, is that the length of the time series, are often one or two magnitudes of order larger than the (sentence-level) NLP problems. Plenty of approaches aim to mitigate the quadratic complexity $O(T^2)$ in the sequence/time series length $T$ to at most $O(T \log T)$. For example, the Informer architecture (Zhou et al., 2020), arguably one of the most accurate forecasting models researched so far, adapts the transformer by a sparse attention mechanism and a successive downsampling/coarsening of the past time series. As in the original transformer, only the coarsest representation is fed into the decoder. Possibly to remedy the loss in resolution by this procedure, the Informer feeds its input a second time into the decoder network, this time without any coarsening.

While forecasting problems share many commonalities with image segmentation problems, transformer-based architectures like the Informer do not involve coupled down- and upscaling procedures to yield predictions on the same resolution as the inputs. Thus, we propose a novel Y-shaped architecture called Yformer that

1. Couples downscaling/upscaling to leverage both, coarse and fine-grained features for time series forecasting,

2. Combines the coupled scaling mechanism with sparse attention modules to capture long-range effects on all scale levels, and

3. Stabilizes encoder and decoder stacks by reconstructing the recent past.

## 2 RELATED WORK

**Deep Learning Based Time Series Forecasting**: While Convolutional Neural Network (CNN) and Recurrent Neural network (RNN) based architectures (Salinas et al., 2020; Rangapuram et al., 2018) outperform traditional methods like ARIMA (Box & Jenkins, 1968) and exponential smoothing methods (Hyndman & Athanasopoulos, 2018), the addition of attention layers (Vaswani et al., 2017) to model time series forecasting has proven to be very beneficial across different problem settings (Fan et al., 2019; Qin et al., 2017; Lai et al., 2018). Attention allows direct pair-wise interaction with eccentric events (like holidays) and can model temporal dynamics inherently unlike RNN's and CNN's that fail to capture long-range dependencies directly. Recent work like Reformer (Kitaev et al., 2020), Linformer (Wang et al., 2020) and Informer (Zhou et al., 2020) have focused on reducing the quadratic complexity of modeling pair-wise interactions to $O(T \log T)$ with the introduction of restricted attention layers. Consequently, they can predict for longer forecasting horizons but are hindered by their capability of aggregating features and maintaining the resolution required for far horizon forecasting.

**U-Net**: The Yformer model is inspired by the famous U-Net architecture introduced in Ronneberger et al. (2015) originating from the field of medical image segmentation. The U-net architecture is capable of compressing information by aggregating over the inputs and up-sampling embeddings to the same resolutions as that of the inputs from their compressed latent features. Current transformer architectures like the Informer (Zhou et al., 2020) do not utilize up-sampling techniques even though the network produces intermediate multi resolution feature maps. Our work aims to capitalize on these multi resolution feature maps and use the U-net shape effectively for the task of time series forecasting. Previous works like Stoller et al. (2019) and Perslev et al. (2019) have successfully applied U-Net architecture for the task of sequence modeling and time series segmentation, illustrating superior results in the respective tasks. These work motivate the use of a U-Net-inspired architecture for time series forecasting as current methods fail to couple sparse attention mechanism with the U-Net shaped architecture. Additional related works section is decoupled from the main text and is presented in the appendix section B.

## 3 PROBLEM FORMULATION

By a **time series** $x$ **with** $M$ **channels**, we mean a finite sequence of vectors in $\mathbb{R}^M$, denote their space by $\mathbb{R}^{*\times M} := \bigcup_{T\in\mathbb{N}} \mathbb{R}^{T\times M}$, and their length by $|x| := T$ (for $x \in \mathbb{R}^{T\times M}$, $M \in \mathbb{N}$). We write $(x, y) \in \mathbb{R}^{*\times(M+O)}$ to denote two time series of same length with $M$ and $O$ channels, respectively.

We model a **time series forecasting instance** as a quadruple $(x, y, x', y') \in \mathbb{R}^{*\times(M+O)} \times \mathbb{R}^{*\times(M+O)}$, where $x, y$ denote the past predictors and targets until a reference time point $T$ and $x', y'$ denote the future predictors and targets from the reference point $T$ to the next $\tau$ time steps. Here, $\tau = |x'|$ is called the forecast horizon.

For a **Time Series Forecasting Problem**, given (i) a sample $\mathcal{D} := \{(x_1, y_1, x'_1, y'_1), \ldots, (x_N, y_N, x'_N, y'_N)\}$ from an unknown distribution $p$ of time series forecasting instances and (ii) a function $\ell : \mathbb{R}^{*\times(O+O)} \to \mathbb{R}$ called loss, we attempt to find a function $\hat{y} : \mathbb{R}^{*\times(M+O)} \times \mathbb{R}^{*\times M} \to \mathbb{R}^{*\times O}$ (with $|\hat{y}(x, y, x')| = |x'|$) with minimal expected loss

$$\mathbb{E}_{(x,y,x',y')\sim p} \, \ell(y', \hat{y}(x, y, x')) \tag{1}$$

The loss $\ell$ usually is the mean absolute error (MAE) or mean squared error (MSE) averaged over future time points:

$$\ell^{\text{mae}}(y', \hat{y}) := \frac{1}{|y'|} \sum_{t=1}^{|y'|} \frac{1}{O} ||y'_t - \hat{y}_t||_1, \quad \ell^{\text{mse}}(y', \hat{y}) := \frac{1}{|y'|} \sum_{t=1}^{|y'|} \frac{1}{O} ||y'_t - \hat{y}_t||_2^2 \tag{2}$$

Furthermore, if there is only one target channel and no predictor channels ($O = 1$, $M = 0$), the time series forecasting problem is called **univariate**, otherwise **multivariate**.

## 4 BACKGROUND

Our work incorporates restricted attention based Transformer in a U-Net inspired architecture. For this reason, we base our work on the current state of the art sparse attention model Informer, introduced in Zhou et al. (2020). We provide a brief overview of the sparse attention mechanism (*ProbSparse*) and the encoder block (*Contracting ProbSparse Self-Attention Blocks*) used in the Informer model for completeness.

***ProbSparse* Attention**: The *ProbSparse* attention mechanism restricts the canonical attention (Vaswani et al., 2017) by selecting a subset $u$ of dominant queries having the largest variance across all the keys. Consequently, the query $\boldsymbol{Q} \in \mathbb{R}^{L_Q \times d}$ in the canonical attention is replaced by a sparse query matrix $\overline{\boldsymbol{Q}} \in \mathbb{R}^{L_Q \times d}$ consisting of only the $u$ dominant queries. *ProbSparse* attention can hence be defined as:

$$\mathcal{A}^{\text{PropSparse}}(\overline{\boldsymbol{Q}}, \boldsymbol{K}, \boldsymbol{V}) = \text{Softmax}(\frac{\overline{\boldsymbol{Q}}\boldsymbol{K}^T}{\sqrt{d}})\boldsymbol{V} \tag{3}$$

where $d$ denotes the input dimension to the attention module. For more details on the *ProbSparse* attention mechanism, we refer the reader to Zhou et al. (2020).

**Contracting ProbSparse Self-Attention Blocks**: The Informer model uses *Contracting ProbSparse Self-Attention Blocks* to distill out redundant information from the long history input sequence $(x, y)$ in a pyramid structure motivated from the image domain (Lin et al., 2017). The sequence of operations within a block begins with a *ProbSparse* self-attention that takes as input the hidden representation $h_i$ from the $i^{th}$ block and projects the hidden representation into query, key and value for self-attention. This is followed by multiple layers of convolution (Conv1d), and finally the MaxPool operation reduces the latent dimension by effectively distilling out redundant information at each block. We refer the reader to Algorithm 2 in the appendix section C where these operations are presented in an algorithmic structure for a comprehensive overview.

# 5 METHODOLOGY

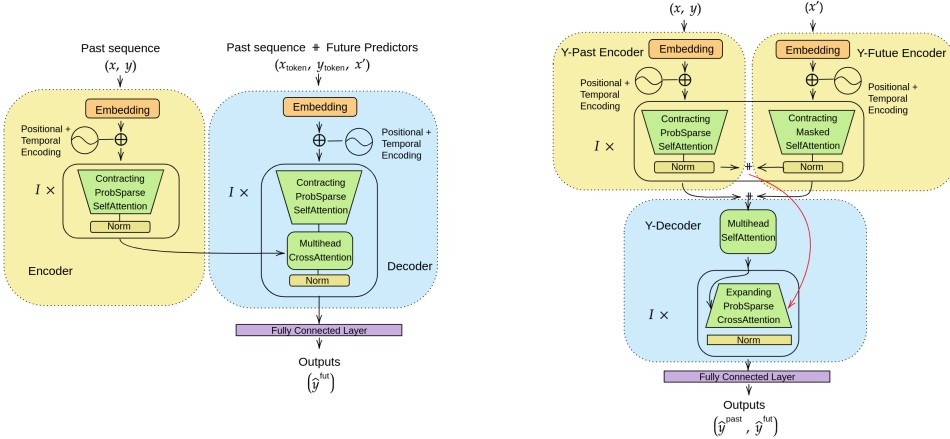

(a) Informer Architecture

(b) Yformer Architecture

Figure 1: Comparison of Informer and Yformer architecture. (1) The Informer architecture process part of the past input data $(x, y)$ within the decoder as $(x_{\text{token}}, y_{\text{token}})$ along with the future predictors $(x')$. The Yformer avoids this redundant reprocessing and uses a masked self-attention network for embedding the future predictors $(x')$. (2) The Informer uses the final encoder embedding as the input to the decoder. The Yformer passes a concatenated ( $+$ ) representation $(e_i)$ of the $i^{\text{th}}$ Y-Past and Y-Future Encoder embedding to the $I - i^{th}$ layer of the Y-Decoder, forming a U-Net connection (represented in red) between the encoder and the decoder. (3) The Yformer architecture predicts both the input reconstruction $\hat{y}^{\text{past}}$ and future predictions $\hat{y}^{\text{fut}}$, to encourage learning embeddings that can produce similar output distribution as the inputs.

The Yformer model is a Y-shaped (Figure 1b) symmetric encoder-decoder architecture that is specifically designed to take advantage of the multi-resolution embeddings generated by the *Contracting ProbSparse Self-Attention Blocks*. The fundamental design consideration is the adoption of U-Net-inspired connections to extract encoder features at multiple resolutions and provide direct connection to the corresponding symmetric decoder block (simple illustration provided in Figure 4, appendix section A). Furthermore, the addition of reconstruction loss helps the model learn generalized embeddings that better approximate the data generating distribution.

The **Y-Past Encoder** of the Yformer is designed using a similar encoder structure as that of the Informer. The Y-Past Encoder embeds the past sequence $(x, y)$ into a scalar projection along with the addition of positional and temporal embeddings. Multiple *Contracting ProbSparse Self-Attention Blocks* are used to generate encoder embeddings at various resolutions. The Informer model uses the final low-dimensional embedding as the input to the decoder (Figure 1a) whereas, the Yformer retains the embeddings at multiple resolutions to be passed on to the decoder. This allows the Yformer to use high-dimensional lower-level embeddings effectively.

The **Y-Future Encoder** of the Yformer mitigates the issue of the redundant reprocessing of parts of the past sequence $(x, y)$ used as tokens $(x_{\text{token}}, y_{\text{token}})$ in the Informer architecture. The Informer model uses only the coarsest representation from the encoder embedding, leading to a loss in resolution and forcing the Informer to pass part of the past sequence as tokens $(x_{\text{token}}, y_{\text{token}})$ to the decoder (Figure 1a). The Yformer separates the future predictors and the past sequence $(x, y)$ bypassing the future predictors $(x')$ through a separate encoder and utilizing the multi-resolution embeddings to dismiss the need for tokens entirely. Unlike the Y-Past Encoder, the attention blocks in the Y-Future encoder are based on masked canonical self-attention mechanism (Vaswani et al., 2017). Masking the attention ensures that there is no information leak from the future time steps to the past. Moreover, a masked canonical self-attention mechanism helps reduce the complexity, as half of the query-key interactions are restricted by design. Thus, the Y-Future Encoder is designed by stacking

multiple *Contracting ProbSparse Self-Attention Blocks* where the *ProbSparse* attention is replaced by the *Masked Attention*. We name these blocks *Contracting Masked Self-Attention Blocks* (Algorithm 3 appendix section C).

The Yformer processes the past inputs and the future predictors separately within its encoders. However, considering the time steps, the future predictors are a continuation of the past time steps. For this reason, the Yformer model concatenates (represented by the symbol $\#$ ) the past encoder embedding and the future encoder embedding along the time dimension after each encoder block, preserving the continuity between the past input time steps and the future time steps. Let $i$ represent the index of an encoder block, then $e_{i+1}^{\text{past}}$ and $e_{i+1}^{\text{fut}}$ represent the output from the past encoder and the future encoder respectively. The final concatenated encoder embedding $(e_{i+1})$ is calculated as,

$$
\begin{aligned}
e_{i+1}^{\text{past}} &= \text{ContractingProbSparseSelfAttentionBlock}(e_i^{\text{past}}) \\
e_{i+1}^{\text{fut}} &= \text{ContractingMaskedSelfAttentionBlock}(e_i^{\text{fut}}) \\
e_{i+1} &= e_{i+1}^{\text{past}} \# e_{i+1}^{\text{fut}}
\end{aligned}
\tag{4}
$$

The encoder embeddings represented by $\mathcal{E} = [e_0, \ldots, e_I]$ (where $I$ is the number of encoder layers) contain the combination of past and future embeddings at multiple resolutions.

The **Y-Decoder** of the Yformer consists of two parts. The first part takes as input the final concatenated low-dimensional embedding $(e_I)$ and performs a multi-head canonical self-attention mechanism. Here, the past encoder embedding $(e_I^{\text{past}})$ is allowed to attend to itself as well as the future encoder embedding $(e_I^{\text{fut}})$ in an unrestricted fashion. The encoder embedding $(e_I)$ is the low-dimensional distilled embedding, and skipping query-key interaction within these low-dimensional embeddings might deny the model useful pair-wise interactions. Therefore, it is by design that this is the only part of the Yformer model that uses canonical self-attention in comparison to the Informer that uses canonical attention within its repeating decoder block, as shown in Figure 1a. Since the canonical self-attention layer is separated from the repeating attention blocks within the decoder, the Yformer complexity from this full attention module does not increase with an increase in the number of decoder blocks. The U-Net architecture inspires the second part of the Y-Decoder. Consequently, the decoder is structured in a symmetric expanding path identical to the contracting encoder. We realize this architecture by introducing upsampling on the ProbSparse attention mechanism using *Expanding ProbSparse Cross-Attention Block*.

The ***Expanding ProbSparse Cross-Attention Block*** within the Yformer decoder performs two tasks: (1) upsample the compressed encoder embedding $e_I$ and (2) perform restricted cross attention between the expanding decoder embedding $d_{I-i}$ and the corresponding encoder embedding $e_i$ (represented in Figure 4 appendix section A). We accomplish both the tasks by introducing an *Expanding ProbSparse Cross-Attention Block* as illustrated in Algorithm 1.

---
**Algorithm 1** Expanding ProbSparse Cross-Attention Block
---
*Input* : $d_{I-i}, e_i$
*Output* : $d_{I-i+1}$
$d_{I-i+1} \leftarrow \text{ProbSparseCrossAttn}(d_{I-i}, e_i)$
$d_{I-i+1} \leftarrow \text{Conv1d}(d_{I-i+1})$
$d_{I-i+1} \leftarrow \text{Conv1d}(d_{I-i+1})$
$d_{I-i+1} \leftarrow \text{LayerNorm}(d_{I-i+1})$
$d_{I-i+1} \leftarrow \text{ELU}(\text{ConvTranspose1d}(d_{I-i+1}))$

---

The *Expanding ProbSparse Cross-Attention Blocks* within the Yformer decoder uses a ProbSparseCrossAttn to construct direct connections between the lower levels of the encoder and the corresponding symmetric higher levels of the decoder. Direct connections from the encoder to the decoder are an essential component for majority of the models within the image domain. For example, ResNet (He et al., 2016), and DenseNet (Huang et al., 2017) have demonstrated that direct connections between previous feature maps, strengthen feature propagation, reduce parameters, mitigate vanishing gradients and encourage feature reuse. However, current transformer-based architectures like the Informer fail to utilize such direct connections. The ProbSparseCrossAttn

takes in as input the decoder embedding from the previous layer $d_{I-i}$ as queries and the corresponding encoder embedding $e_i$ as keys. The Yformer uses the *ProbSparse* restricted attention so that the model is scalable with an increase in the number of decoder blocks.

We utilize $\mathrm{ConvTranspose1d}$ or popularly known as $\mathrm{Deconvolution}$ for incrementally increasing the embedding space. The famous U-Net architecture uses a symmetric expanding path using such $\mathrm{Deconvolution}$ layers. This property enables the model to not only aggregate over the input but also upscale the latent dimensions, improving the overall expressivity of the architecture. The decoder of Yformer follows a similar strategy by employing $\mathrm{Deconvolution}$ to expand the embedding space of the encoded output. We describe the different operators used in the appendix section C.

A fully connected layer ($\mathrm{LinearLayer}$) predicts the future time steps $\hat{y}^{\mathrm{fut}}$ from the final decoder layer ($d_I$) and additionally reconstructs the past input targets $\hat{y}^{\mathrm{past}}$.

$$[\hat{y}^{\mathrm{past}}, \hat{y}^{\mathrm{fut}}] = \mathrm{LinearLayer}(d_I) \tag{5}$$

The addition of reconstruction loss to the Yformer as an auxiliary loss, serves two significant purposes. Firstly, the reconstruction loss acts as a data-dependent regularization term that reduces overfitting by learning embeddings that are more general (Ghasedi Dizaji et al., 2017; Jarrett & van der Schaar, 2020). Secondly, the reconstruction loss helps in producing future-output in a similar distribution as the inputs (Bank et al., 2020). For far horizon forecasting, we are interested in learning a future-output distribution. However, the future-output distribution and the past-input distribution arise from the same data generating process. Therefore having an auxiliary reconstruction loss would direct the gradients to a better approximate of the data generating process. The Yformer model is trained on the combined loss $\ell$,

$$\ell = \alpha\,\ell^{\mathrm{mse}}(y, \hat{y}^{\mathrm{past}}) + (1 - \alpha)\,\ell^{\mathrm{mse}}(y', \hat{y}^{\mathrm{fut}}) \tag{6}$$

where the first term tries to learn the past targets $y$ and the second term learns the future targets $y'$. We use the reconstruction factor ($\alpha$) to vary the importance of reconstruction and future prediction and tune this as a hyperparameter.

## 6 EXPERIMENTS

### 6.1 DATASETS

To evaluate our proposed YFormer architecture, we use two real-world public datasets and one public benchmark to compare the experiment results with the published results by the Informer.

**ETT** (Electricity Transformer Temperature[1]): This real-world dataset for the electric power deployment introduced by Zhou et al. 2020 combines short-term periodical patterns, long-term periodical patterns, long-term trends, and irregular patterns. The data consists of load and temperature readings from two transformers and is split into two hourly subsets ETTh1 and ETTh2. The ETTm1 dataset is generated by splitting ETTh1 dataset into 15-minute intervals. The dataset has six features and 70,080 data points in total. For easy comparison, we kept the splits for train/val/test consistent with the published results in Zhou et al. 2020, where the available 20 months of data is split as 12/4/4.

**ECL** (Electricity Consuming Load[2]): This electricity dataset represents the electricity consumption from 2011 to 2014 of 370 clients recorded in 15-minutes periods in Kilowatt (kW). We split the data into 15/3/4 months for train, validation, and test respectively as in Zhou et al. 2020.

### 6.2 EXPERIMENTAL SETUP

**Baseline**: Our main baseline is the the Informer architecture by Zhou et al. 2020. The results presented in the paper were reported to have a data scaling issue[3], and the authors have updated

---

[1]Available under https:// github.com/zhouhaoyi/ETDataset.

[2]Available under https://archive.ics.uci.edu/ml/ datasets/ElectricityLoadDiagrams20112014

[3]https://github.com/zhouhaoyi/Informer2020/issues/41

Table 1: MAE for the **univariate** time series forecasting task. The best result is highlighted in bold and the second-best in italic and red. Informer* here represents a modified version of the standard informer, which uses the canonical attention module.

| Dataset | Horizon ($\tau$) | Methods | | | | | Improvement % |
|---------|------------------|---------|--------|----------|----------|---------|---------------|
|         |                  | LogTrans | LSTnet | Informer* | Informer | Yformer |               |
| ETTh1   | 24  | 0.259 | 0.280 | *0.246* | 0.247 | **0.230** | 6.50 |
|         | 48  | 0.328 | 0.327 | 0.322 | *0.319* | **0.308** | 3.45 |
|         | 168 | 0.375 | 0.422 | 0.355 | *0.346* | **0.268** | 22.54 |
|         | 336 | 0.398 | 0.552 | *0.369* | 0.387 | **0.365** | 1.08 |
|         | 720 | 0.463 | 0.707 | *0.421* | 0.435 | **0.394** | 6.41 |
| ETTh2   | 24  | 0.255 | 0.263 | 0.241 | *0.240* | **0.221** | 7.92 |
|         | 48  | 0.348 | 0.341 | *0.317* | **0.314** | 0.334 | −6.37 |
|         | 168 | 0.422 | 0.414 | 0.390 | *0.389* | **0.337** | 13.37 |
|         | 336 | 0.437 | 0.607 | 0.423 | *0.417* | **0.391** | 6.24 |
|         | 720 | 0.493 | 0.58  | 0.442 | *0.431* | **0.382** | 11.37 |
| ETTm1   | 24  | 0.202 | 0.243 | 0.160 | *0.137* | **0.118** | 13.87 |
|         | 48  | 0.220 | 0.362 | *0.194* | 0.203 | **0.173** | 10.82 |
|         | 96  | 0.386 | 0.496 | 0.384 | *0.372* | **0.311** | 16.40 |
|         | 288 | 0.572 | 0.795 | *0.548* | 0.554 | **0.316** | 42.34 |
|         | 672 | 0.702 | 1.352 | 0.664 | *0.644* | **0.476** | 26.09 |
| ECL     | 48  | 0.429 | *0.357* | 0.368 | 0.359 | **0.322** | 9.80 |
|         | 168 | 0.529 | *0.436* | 0.514 | 0.503 | **0.361** | 17.20 |
|         | 336 | 0.563 | *0.519* | 0.552 | 0.528 | **0.375** | 27.75 |
|         | 720 | 0.609 | 0.595 | 0.578 | *0.571* | **0.479** | 19.50 |
|         | 960 | 0.645 | 0.683 | 0.638 | *0.608* | **0.573** | 16.11 |
| # wins per method | | 0 | 0 | 0 | 1 | 19 | |
|         |     |       |       |       |       | Avg | 13.62 |

their results in the official GitHub repository[4]. Therefore we compare against these updated results. Recently, the Query Selector (Klimek et al., 2021) utilize the Informer framework for far horizon forecasting, however they report results from the Informer paper before the bug fix and hence we avoid comparison with this work. As a second baseline, we also compare the second-best performing model in Zhou et al. 2020 which is the Informer that uses canonical attention module. It is represented as Informer* in the tables. Furthermore, we also compare against DeepAR (Salinas et al., 2020), LogTrans (Li et al., 2019), and LSTnet (Lai et al., 2018) architectures as they outperformed the Informer baseline for certain forecasting horizons. For a quick analysis, we present the percent improvement achieved by the Yformer over the current best results.

For easy comparison, we choose two commonly used metrics for time series forecasting to evaluate the Yformer architecture, the MAE and MSE in Equation 2. We report the results for the MAE in the paper and provide the MSE results in the appendix section D. We performed our experiments on GeForce RTX 2080 Ti GPU nodes with 32 GB ram and provide results as an average of three runs.

## 6.3 RESULTS AND ANALYSIS

This section compares our results with the results reported in the Informer baseline both in uni- and multivariate settings for the multiple datasets and horizons. The best-performing and the second-best models (lowest MAE) are highlighted in bold and red, respectively.

**Univariate**: The proposed Yformer model is able to outperform the current state of the art Informer baseline in 19 out of the 20 available tasks across different datasets and horizons by an average of 13.62 % of MAE. The Table 1 illustrates that the superiority of the Yformer is not just limited to a far horizon but even for the shorter horizons and in general across datasets. Considering the individual datasets, the Yformer surpasses the current state of the art by 8, 6.8, 21.9, and 18.1% of MAE for the ETTh1, ETTh2, ETTm1, and ECL datasets respectively as seen in Table 1. We also report MSE scores in the supplementary appendix section D, which illustrates an improvement of 16.7, 12.6, 34.8 and 15.2% for the ETTh1, ETTh2, ETTm1, and ECL datasets respectively. We observe that the MAE for the model is greater at horizon 48 than the MAE at horizon 168 for the ETTh1 dataset. This may be a case where the reused hyperparameters from the Informer paper are far from optimal for

---

[4]Commit 702fb9bbc69847ecb84a8bb205693089efb41c6

Table 2: MAE for the **multivariate** time series forecasting task. The best result is highlighted in bold and the second-best in italic and red. Informer* here represents a modified version of the standard informer, which uses the canonical attention module.

| Dataset | Horizon ($\tau$) | Methods | | | | | Improvement % |
|---|---|---|---|---|---|---|---|
| | | LogTrans | LSTnet | Informer* | Informer | Yformer | |
| ETTh1 | 24 | 0.604 | 0.901 | 0.577 | *0.549* | **0.492** | 10.38 |
| | 48 | 0.757 | 0.960 | 0.671 | *0.625* | **0.537** | 14.08 |
| | 168 | 0.846 | 1.214 | 0.797 | *0.752* | **0.684** | 9.04 |
| | 336 | 0.952 | 1.369 | *0.813* | 0.873 | **0.803** | 1.23 |
| | 720 | 1.291 | 1.380 | 0.917 | *0.896* | **0.803** | 10.38 |
| ETTh2 | 24 | 0.750 | 1.457 | 0.727 | *0.665* | **0.498** | 25.11 |
| | 48 | 1.034 | 1.687 | 1.077 | *1.001* | **0.865** | 13.59 |
| | 168 | 1.681 | 2.513 | 1.612 | *1.515* | **1.218** | 19.60 |
| | 336 | 1.763 | 2.591 | *1.285* | 1.340 | **1.283** | 0.16 |
| | 720 | 1.552 | 3.709 | 1.495 | *1.473* | **1.337** | 9.23 |
| ETTm1 | 24 | 0.412 | 1.170 | 0.371 | *0.369* | **0.363** | 1.63 |
| | 48 | 0.583 | 1.215 | *0.470* | 0.503 | **0.457** | 2.77 |
| | 96 | 0.792 | 1.542 | *0.612* | 0.614 | **0.567** | 7.35 |
| | 288 | 1.320 | 2.076 | 0.879 | *0.786* | **0.593** | 24.55 |
| | 672 | 1.461 | 2.941 | 1.103 | *0.926* | **0.656** | 29.16 |
| ECL | 48 | 0.418 | 0.445 | 0.399 | *0.393* | **0.390** | 0.76 |
| | 168 | 0.432 | 0.476 | *0.420* | 0.424 | **0.387** | 7.86 |
| | 336 | 0.439 | 0.477 | 0.439 | *0.431* | **0.394** | 8.58 |
| | 720 | 0.454 | 0.565 | *0.438* | 0.443 | **0.384** | 12.33 |
| | 960 | 0.589 | 0.599 | 0.550 | *0.548* | **0.388** | 29.20 |
| # wins per method | 0 | 0 | 0 | 0 | 0 | 20 | |
| | | | | | | Avg | 11.85 |

the Yformer. The other results show consistent behavior of increasing error with increasing horizon length $\tau$. Additionally, this behavior is also observed in the Informer baseline for ETTh2 dataset (Table 2), where the loss is 1.340 for horizon 336 and 1.515 for a horizon of 168.

Figure 2 differentiates the improvement of Y-former architecture from the baseline Informer and the reconstruction loss. We notice that the Yformer architecture without the reconstruction loss is able to outperform the Informer architecture across different datasets and horizons. Additionally, the optimal value for the loss weighting hyperparameter $\alpha$ is always greater than zero as shown in Table 3, confirming the assumption that the addition of the auxiliary reconstruction loss term helps the model to generalize well to the future distribution.

**Multivariate**: We observe a similar trend in the multivariate setting. Here the Yformer model outperforms the current state of the art method in all of the 20 tasks across the four datasets by a margin of 11.85% of MAE. There is a clear superiority of the proposed approach in longer horizons across the different datasets. Across the different datasets, the Yformer improves the current state of the art MAE results by 9, 13.5, 13.1, and 11.7% of MAE and 12, 26.3, 13.9, and 17.1% of MSE for the ETTh1, ETTh2, ETTm1, and ECL datasets respectively (table in the supplementary appendix section D). We attribute the improvement in performance to superior architecture and the ability to approximate the data distribution for the multiple targets due to the reconstruction loss.

## 7 ABLATION STUDY

We performed additional experiments on the ETTh2 and ETTm1 datasets to study the impact of the Yformer model architecture and the effect of the reconstruction loss separately.

### 7.1 Y-FORMER ARCHITECTURE

In this section we attempt to understand (1) How much of an improvement is brought about by the Y-shaped model architecture? and (2) How much impact did the reconstruction loss have? From Figure 2, it is clear that the Yformer architecture performs better or is comparable to the Informer throughout the entire horizon range (without the reconstruction loss $\alpha = 0$). Moreover, we can notice that for the larger horizons, the Yformer architecture (with $\alpha = 0$) seems to have a clear

| $\alpha$ | Horizon $\tau$ | | | count |
|---|---|---|---|---|
| | short | medium | long | |
| 0 | 0 | 0 | 1 | 1 |
| 0.1 | 0 | 1 | 2 | 3 |
| 0.3 | 1 | 2 | 1 | 4 |
| 0.5 | 1 | 0 | 0 | 1 |
| 0.7 | 5 | 2 | 3 | 10 |
| 1 | 0 | 0 | 0 | 0 |

| $\alpha$ | Horizon $\tau$ | | | count |
|---|---|---|---|---|
| | short | medium | long | |
| 0 | 1 | 0 | 1 | 2 |
| 0.1 | 0 | 0 | 1 | 1 |
| 0.3 | 1 | 0 | 1 | 2 |
| 0.5 | 1 | 1 | 1 | 3 |
| 0.7 | 5 | 4 | 2 | 11 |
| 1 | 0 | 0 | 1 | 1 |

Table 3: Impact of the $\alpha$ parameter on univariate (top) and multivariate (bottom) problem settings for the Yformer architecture. The horizons $\tau$ are 0-50, 51-300, and $> 300$ for short, medium, and long horizons respectively.

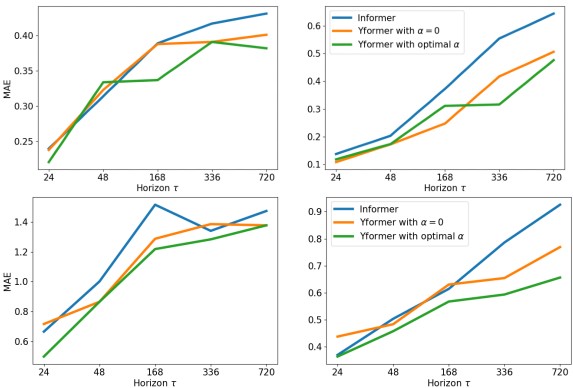

Figure 2: Impact of the Yformer architecture and reconstruction loss on univariate (top) and multivariate (bottom) for ETTh2 (left) and ETTm1 (right) datasets. X-axis shows the forecast horizons ($\tau$), Y-axis shows the MAE. The colours blue, orange, green represents the Informer, the Yformer without reconstruction loss, the Yformer respectively.

advantage over in Informer baseline. We attribute this improvement in performance to the direct U-Net inspired connections within the Yformer architecture. Using feature maps at multiple resolutions offers a clear advantage by eliminating vanishing gradients and encouraging feature reuse.

The reconstruction loss seems to have a significant impact in reducing the loss for the ETTm1 multivariate case (right bottom graph Figure 2). And, in general, it helps to improve the results for the Yformer architecture as the green curve in the Figure 2 is almost always below the blue (Informer) and orange (Yformer with $\alpha = 0$) curve. The significance of the loss depends on the dataset but, in general, the auxiliary loss helps to improve the overall performance.

## 7.2 RECONSTRUCTION FACTOR

How impactful is the reconstruction factor $\alpha$ from the proposed loss in Eq. 6? We analyzed the optimal value for $\alpha$ across different forecasting horizons and summarized them in Table 3. Interestingly, $\alpha = 0.7$ seems to be the predominant optimal setting, implying a high weight for the reconstruction loss helps the Yformer to achieve a lower loss for the future targets. Additionally, we can observe a trend that $\alpha$ is on average larger for short forecasting horizons signifying that the input reconstruction loss is also important for the short horizon forecast.

## 8 CONCLUSION

Time series forecasting is an important business and research problem that has a broad impact in today's world. This paper proposes a novel Y-shaped architecture, specifically designed for the far horizon time series forecasting problem. The study shows the importance of direct connections from the multi-resolution encoder to the decoder and reconstruction loss for the task of time series forecasting. The Yformer couples the U-Net architecture from the image segmentation domain on a sparse transformer model to achieve state of the art results in 39 out of the 40 tasks across various settings as presented in the Tables 1 and 2.

## 9 REPRODUCIBILITY STATEMENT

All the experimental datasets used to evaluate the Yformer architecture are publicly available and, we provide the source code for the proposed architecture with the supplementary materials for reproducibility. The hyperparameters needed to reproduce the reported results on the different datasets are presented in the appendix section E.

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

# A APPENDIX : ADDITIONAL FIGURES

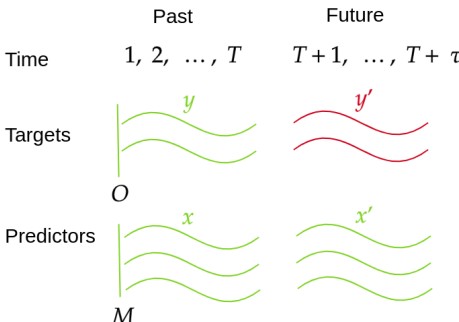

Figure 3: General time series setting illustrating the quadruples $(x, y, x', y')$ denoting the *past predictors*, *past targets*, *future predictors* and *future targets* respectively. Given the history information $(x, y)$ and the future predictors $(x')$ (in green) until time $t = T$, time series forecasting predicts the target $y'$ (in red) for the next $\tau$ time steps. In the figure, $O$ and $M$ represents the respective channels of the targets and the predictors.

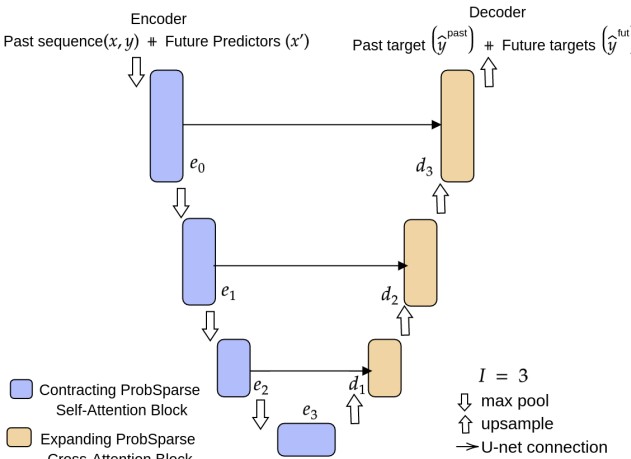

Figure 4: U-Net connections for effectively utilizing embeddings at multiple resolutions in the Yformer. The Y-Past Encoder embeddings and the Y-Future Encoder embeddings are concatenated within the Yformer encoder. A direct connection is allowed between the contracting encoder embedding $(e_i)$ and the corresponding expanding decoder embedding $(d_{I-i})$. ($+\!\!\!+$ denotes concatenation)

# B APPENDIX : ADDITIONAL RELATED WORKS

**Time-series Forecasting**: Time Series forecasting has been a well-established research topic with steadily growing applications in a real-world environment. Traditionally, ARIMA Box & Jenkins (1968) and exponential smoothing methods Hyndman & Athanasopoulos (2018) have been used for time series forecasting. However, limited scalability and the inability to model non-linear patterns restrict their use for time series forecasting.

**Reconstruction Loss** The usage of a reconstruction loss originated in the domain of variational autoencoders Kingma & Welling (2013) to reconstruct the inputs during the generative process. Forecasting for a long horizon in a single forward pass can be considered as generating a distribution given the past time steps as the input distribution. A reconstruction loss would hence be beneficial as

the forecast distribution is similar to the past input distribution. Moreover, a recent study by Le et al. 2018 has shown that the addition of the reconstruction term to any loss function generally provides uniform stability and bounds on the generalization error, therefore leading to a more robust model overall with no negative effect on the performance.

Our work tries to combine the different ideas of U-Net from image segmentation with a sparse attention network for modeling time series that also utilizes the additional guidance from reconstruction loss.

## C  APPENDIX : DEFINITION

### C.1  OPERATORS

**ProbSparseAttn**: Attention module that uses the ProbSparse method introduced in Zhou et al. (2020). The query matrix $\overline{Q} \in \mathbb{R}^{L_Q \times d}$ denotes the sparse query matrix with $u$ dominant queries.

$$\mathcal{A}^{\text{PropSparse}}(\overline{Q}, K, V) = \text{Softmax}(\frac{\overline{Q}K^T}{\sqrt{d}})V \tag{7}$$

**MaskedAttn**: Canonical self-attention with masking to prevent positions from attending to subsequent positions in the future (Vaswani et al., 2017).

**Conv1d**: Given $N$ batches of 1D array of length $L$ and $C$ number of channels/dimensions. A convolution operation produces an output:

$$\text{out}(N_i, C_{\text{out}_j}) = \text{bias}(C_{\text{out}_j}) + \sum_{k=0}^{C_{in}-1} \text{weight}(C_{\text{out}_j}, k) \star \text{input}(N_i, k) \tag{8}$$

For further reference please visit pytorch Conv1D page

**LayerNorm**: Layer Normalization introduced in Ba et al. (2016), normalizes the inputs across channels/dimensions. $\text{LayerNorm}$ is the default normalization in common transformer architectures (Vaswani et al., 2017). Here, $\gamma$ and $\beta$ are learnable affine transformations.

$$\text{out}(N, *) = \frac{\text{input}(N, *) - \text{E}[\text{input}(N, *)]}{\sqrt{\text{Var}[\text{input}(N, *)] + \epsilon}} * \gamma + \beta \tag{9}$$

**MaxPool**: Given $N$ batches of 1D array of length $L$, and $C$ number of channels/dimensions. A $\text{MaxPool}$ operation produces an output.

$$\text{out}(N_i, C_j, k) = \max_{m=0,\ldots,\text{kernel\_size}-1} \text{input}(N_i, C_j, \text{stride} \times k + m) \tag{10}$$

For further reference please visit pytorch MaxPool1D page

**ELU**: Given an input $x$, the ELU applies element-wise non linear activation function as shown.

$$\text{ELU}(x) = \begin{cases} x, & \text{if } x > 0 \\ \alpha * (\exp(x) - 1), & \text{if } x \leq 0 \end{cases} \tag{11}$$

**ConvTranspose1d**: Also known as deconvolution or fractionally strided convolution, uses convolution on padded input to produce upsampled outputs (see pytorch ConvTranspose1d page).

### C.2  CONTRACTING PROBSPARSE SELF-ATTENTION BLOCKS

The Informer model uses *Contracting ProbSparse Self-Attention Blocks* to distil out redundant information from the long history input sequence $(x, y)$ in a pyramid structure similar to Lin et al.

(2017). The sequence of operations within a block begins with a *ProbSparse* self-attention that takes as input the hidden representation $h_i$ from the $i^{th}$ block and projects the hidden representation into query, key and value for self-attention. This is followed by multiple layers of convolution ($\mathrm{Conv1d}$), and finally a $\mathrm{MaxPool}$ operation is performed to reduce the latent dimension at each block, after applying non-linearity with an $\mathrm{ELU}()$ activation function . The $\mathrm{LayerNorm}$ operation, regularizes the model using Layer Normalization (Ba et al., 2016). Algorithm 2 shows the multiple operations performed within a *Contracting ProbSparse Self-Attention Block* that takes an input $h_i$ and produces the hidden representation $h_{i+1}$ for the $i + 1^{th}$ block.

---

**Algorithm 2** Contracting ProbSparse Self-Attention Block

---

*Input* : $h_i$
*Output* : $h_{i+1}$
$h_{i+1} \leftarrow \mathrm{ProbSparseAttn}(h_i, h_i)$
$h_{i+1} \leftarrow \mathrm{Conv1d}(h_{i+1})$
$h_{i+1} \leftarrow \mathrm{Conv1d}(h_{i+1})$
$h_{i+1} \leftarrow \mathrm{LayerNorm}(h_{i+1})$
$h_{i+1} \leftarrow \mathrm{MaxPool}(\mathrm{ELU}(\mathrm{Conv1d}(h_{i+1})))$

---

### C.3 CONTRACTING MASKED SELF-ATTENTION BLOCKS

The Y-Future Encoder, uses multiple blocks of *Contracting Masked Self-Attention Blocks* that replaces the ProbSparseAttn in the *Contracting ProbSparse Self-Attention Blocks* with a masked attention. Masking attention for the Y-Future Encoder, avoids any future information leak architecturally. In our experiments, the addition of restricted attention like the *ProbSparse* on an already masked attention resulted in performance loss. This could be the result of missing query-key interaction brought about by the *ProbSparse* on an already masked attention. Algorithm 3 shows the multiple operations performed within a *Contracting Masked Self-Attention Block* that takes an input $h_i$ and produces the hidden representation $h_{i+1}$ for the $i + 1^{th}$ block.

---

**Algorithm 3** Contracting Masked Self-Attention Block

---

*Input* : $h_i$
*Output* : $h_{i+1}$
$h_{i+1} \leftarrow \mathrm{MaskedAttn}(h_i, h_i)$
$h_{i+1} \leftarrow \mathrm{Conv1d}(h_{i+1})$
$h_{i+1} \leftarrow \mathrm{Conv1d}(h_{i+1})$
$h_{i+1} \leftarrow \mathrm{LayerNorm}(h_{i+1})$
$h_{i+1} \leftarrow \mathrm{MaxPool}(\mathrm{ELU}(\mathrm{Conv1d}(h_{i+1})))$

---

# D   APPENDIX : MSE RESULTS

Table 4: MSE for the **univariate** time series forecasting task with the Yformer architecture. The best result is highlighted in bold and the second best in italic and red. Informer* here is representing a modified version of the standard informer which is using the canonical attention module.

| Dataset | Horizon ($\tau$) | Methods | | | | | Improvement % |
|---|---|---|---|---|---|---|---|
| | | LogTrans | LSTnet | Informer* | Informer | Yformer | |
| ETTH1 | 24 | 0.103 | 0.107 | *0.092* | 0.098 | **0.082** | 10.87 |
| | 48 | 0.167 | 0.162 | 0.161 | *0.158* | **0.139** | 12.03 |
| | 168 | 0.207 | 0.239 | 0.187 | *0.183* | **0.111** | 39.34 |
| | 336 | 0.230 | 0.445 | *0.215* | 0.222 | **0.195** | 9.30 |
| | 720 | 0.273 | 0.658 | *0.257* | 0.269 | **0.226** | 12.06 |
| ETTH2 | 24 | 0.102 | 0.098 | 0.099 | *0.093* | **0.082** | 11.83 |
| | 48 | 0.169 | 0.163 | *0.159* | **0.155** | 0.172 | −10.97 |
| | 168 | 0.246 | 0.255 | 0.235 | *0.232* | **0.174** | 25.00 |
| | 336 | 0.267 | 0.604 | *0.258* | 0.263 | **0.224** | 13.18 |
| | 720 | 0.303 | 0.429 | 0.285 | *0.277* | **0.211** | 23.83 |
| ETTm1 | 24 | 0.065 | 0.091 | 0.034 | *0.030* | **0.024** | 20.00 |
| | 48 | 0.078 | 0.219 | *0.066* | 0.069 | **0.048** | 27.27 |
| | 96 | 0.199 | 0.364 | *0.187* | 0.194 | **0.143** | 23.53 |
| | 288 | 0.411 | 0.948 | 0.409 | *0.401* | **0.150** | 62.59 |
| | 672 | 0.598 | 2.437 | 0.519 | *0.512* | **0.305** | 40.43 |
| ECL | 48 | 0.280 | *0.204* | 0.238 | 0.239 | **0.194** | 4.90 |
| | 168 | 0.454 | *0.315* | 0.442 | 0.447 | **0.260** | 17.46 |
| | 336 | 0.514 | *0.414* | 0.501 | 0.489 | **0.269** | 35.02 |
| | 720 | 0.558 | 0.563 | 0.543 | *0.540* | **0.427** | 20.93 |
| | 960 | 0.624 | 0.657 | 0.594 | **0.582** | *0.595* | −2.23 |
| # wins per method | | 0 | 0 | 0 | 0 | 2 | 18 |
| | | | | | | Avg | 19.82 |

Table 5: MSE for the **multivariate** time series forecasting task with the Yformer architecture. The best result is highlighted in bold and the second best in italic and red. Informer* here is representing a modified version of the standard informer which is using the canonical attention module.

| Dataset | Horizon ($\tau$) | Methods | | | | | Improvement % |
|---|---|---|---|---|---|---|---|
| | | LogTrans | LSTnet | Informer* | Informer | Yformer | |
| ETTH1 | 24 | 0.686 | 1.293 | 0.620 | *0.577* | **0.485** | 15.94 |
| | 48 | 0.766 | 1.456 | 0.692 | *0.685* | **0.530** | 22.63 |
| | 168 | 1.002 | 1.997 | 0.947 | *0.931* | **0.866** | 6.98 |
| | 336 | 1.362 | 2.655 | *1.094* | 1.128 | **1.041** | 4.84 |
| | 720 | 1.397 | 2.143 | 1.241 | *1.215* | **1.098** | 9.63 |
| ETTH2 | 24 | 0.828 | 2.742 | 0.753 | *0.72* | **0.412** | 42.78 |
| | 48 | 1.806 | 3.567 | 1.461 | *1.457* | **1.171** | 19.63 |
| | 168 | 4.070 | *3.242* | 3.485 | 3.489 | **2.171** | 33.04 |
| | 336 | 3.875 | *2.544* | 2.626 | 2.723 | **2.260** | 11.16 |
| | 720 | 3.913 | 4.625 | 3.548 | *3.467* | **2.595** | 25.15 |
| ETTm1 | 24 | 0.419 | 1.968 | *0.306* | 0.323 | **0.289** | 5.56 |
| | 48 | 0.507 | 1.999 | **0.465** | 0.494 | *0.486* | −4.52 |
| | 96 | 0.768 | 2.762 | 0.681 | *0.678* | **0.569** | 16.08 |
| | 288 | 1.462 | 1.257 | 1.162 | *1.056* | **0.649** | 38.54 |
| | 672 | 1.669 | 1.917 | 1.231 | *1.192* | **0.772** | 35.23 |
| ECL | 48 | 0.355 | 0.369 | *0.334* | 0.344 | **0.306** | 8.38 |
| | 168 | 0.368 | 0.394 | *0.353* | 0.368 | **0.317** | 10.20 |
| | 336 | 0.373 | 0.419 | 0.381 | *0.381* | **0.323** | 15.22 |
| | 720 | 0.409 | 0.556 | *0.391* | 0.406 | **0.312** | 20.20 |
| | 960 | 0.477 | 0.605 | 0.492 | *0.460* | **0.315** | 31.52 |
| # wins per method | | 0 | 0 | 0 | 1 | 0 | 19 |
| | | | | | | Avg | 18.41 |

# E  APPENDIX : HYPERPARAMETERS

## E.1  HYPERPARAMETER SEARCH SPACE

For a fair comparison, we retain the design choices from the Informer baseline like the history input length $(T)$ for a particular forecast length $(\tau)$, so that any performance improvement can exclusively be attributed to the architecture of the Yformer model and not to an increased history input length. We performed a grid search for learning rates of $\{0.001, 0.0001\}$, $\alpha$-values of $\{0, 0.3, 0.5, 0.7, 1\}$, number of encoder and decoder blocks $I = \{2, 3, 4\}$ while keeping all the other hyperparameters the same as the Informer. Furthermore, Adam optimizer was used for all experiments, and we employed an early stopping criterion with a patience of three epochs. To counteract overfitting, we tried dropout with varying ratios but interestingly found the effect to be minimal in the results. Therefore, we adopt weight-decay for our experiments with factors $\{0, 0.02, 0.05\}$ for additional regularization. We select the optimal hyperparameters based on the lowest validation loss and will publish the code upon acceptance.

## E.2  OPTIMAL HYPERPARAMETERS

Table 6: Optimal hyperparameters across different horizon and datasets for the univariate setting. All the remaining hyperparameters are retained from the Informer Model.

| Dataset | Horizon $\tau$ | History Length | Weight Decay | Learning Rate | Reconstruction Factor $\alpha$ | Batch Size | Encoder Blocks |
|---|---|---|---|---|---|---|---|
| ETTh1 | 24 | 720 | 0 | 0.0001 | 0.7 | 32 | 2 |
| | 48 | 720 | 0 | 0.0001 | 0.7 | 16 | 4 |
| | 168 | 720 | 0 | 0.001 | 0.7 | 32 | 4 |
| | 336 | 720 | 0.05 | 0.0001 | 0.1 | 32 | 4 |
| | 720 | 720 | 0.05 | 0.0001 | 0.7 | 16 | 2 |
| ETTh2 | 24 | 48 | 0 | 0.0001 | 0.7 | 32 | 2 |
| | 48 | 96 | 0.02 | 0.0001 | 0.3 | 32 | 4 |
| | 168 | 336 | 0.02 | 0.001 | 0.3 | 32 | 2 |
| | 336 | 336 | 0.09 | 0.0001 | 0 | 32 | 2 |
| | 720 | 336 | 0.09 | 0.0001 | 0.7 | 16 | 2 |
| ETTm1 | 24 | 96 | 0.02 | 0.0001 | 0.7 | 32 | 4 |
| | 48 | 96 | 0.02 | 0.0001 | 0.7 | 32 | 4 |
| | 96 | 384 | 0.02 | 0.0001 | 0.1 | 32 | 4 |
| | 288 | 384 | 0.02 | 0.001 | 0.7 | 16 | 2 |
| | 672 | 384 | 0.07 | 0.001 | 0.3 | 16 | 2 |
| ECL | 48 | 168 | 0 | 0.0001 | 0.7 | 16 | 2 |
| | 168 | 168 | 0.01 | 0.0001 | 0.3 | 16 | 2 |
| | 336 | 168 | 0.01 | 0.0001 | 0.7 | 16 | 2 |
| | 720 | 168 | 0 | 0.0001 | 0.1 | 16 | 2 |
| | 960 | 48 | 0 | 0.0001 | 0.5 | 16 | 4 |

Table 7: Optimal hyperparameters across different horizon and datasets for the multivariate setting. All the remaining hyperparameters are retained from the Informer Model.

| Dataset | Horizon $\tau$ | History Length | Weight Decay | Learning Rate | Reconstruction Factor $\alpha$ | Batch Size | Encoder Blocks |
|---|---|---|---|---|---|---|---|
| ETTh1 | 24 | 48 | 0 | 0.0001 | 0.7 | 32 | 3 |
| | 48 | 96 | 0.02 | 0.001 | 0.5 | 32 | 2 |
| | 168 | 168 | 0.02 | 0.001 | 0.7 | 32 | 2 |
| | 336 | 168 | 0 | 0.0001 | 0.7 | 32 | 4 |
| | 720 | 336 | 0.05 | 0.0001 | 1 | 16 | 2 |
| ETTh2 | 24 | 48 | 0 | 0.0001 | 0.7 | 32 | 2 |
| | 48 | 96 | 0.02 | 0.001 | 0 | 32 | 4 |
| | 168 | 336 | 0.09 | 0.001 | 0.7 | 32 | 2 |
| | 336 | 336 | 0.07 | 0.001 | 0.3 | 32 | 2 |
| | 720 | 336 | 0 | 0.0001 | 0 | 16 | 2 |
| ETTm1 | 24 | 672 | 0 | 0.0001 | 0.7 | 32 | 2 |
| | 48 | 96 | 0 | 0.0001 | 0.7 | 32 | 4 |
| | 96 | 384 | 0.05 | 0.0001 | 0.7 | 32 | 4 |
| | 288 | 672 | 0.02 | 0.001 | 0.5 | 16 | 2 |
| | 672 | 672 | 0.02 | 0.0001 | 0.3 | 16 | 2 |
| ECL | 48 | 24 | 0 | 0.0001 | 0.7 | 16 | 3 |
| | 168 | 48 | 0 | 0.0001 | 0.7 | 16 | 3 |
| | 336 | 24 | 0 | 0.0001 | 0.5 | 16 | 2 |
| | 720 | 48 | 0 | 0.0001 | 0.7 | 16 | 2 |
| | 960 | 336 | 0 | 0.0001 | 0.7 | 16 | 2 |

