# OpenReview forum: "Yformer: U-Net Inspired Transformer Architecture for Far Horizon Time Series Forecasting"
_ICLR.cc/2022/Conference — ICLR 2022 Submitted_

### Official Review · Reviewer_HmTV · 2021-10-25

**Correctness:** 2
**Technical Novelty And Significance:** 2
**Empirical Novelty And Significance:** 2
**Recommendation:** 3
**Confidence:** 4

**Main Review:**

-	While the proposed methods hold great promise, my biggest concern is that the experiments seem like unfair comparisons. The authors compare the performance of Yformer against the excerpted results from the Informer. However, in my view, the Yformer and Informer use different problem formulations. According to the authors’ problem formulation, the Yformer predicts the future targets y’ based on the three inputs: past predictors x, past targets y, and future predictors x’. On the other hand, the Informer does not rely on the future predictors x’. The claimed performance improvement by the Yformer could be due to additional information within the future predictors. Furthermore, I’m not sure whether the authors’ problem formulation is appropriate in the real-world setting. Future predictors such as power load features in the ETT dataset would not be the “known” variables for the prediction.
-	While, in the abstract, the authors stated that they used four benchmark datasets, the manuscript only contains experiment results for the three benchmark datasets. Compared to the Informer paper, it seems the results for the Weather dataset are missing. If there are no particular reasons for the exclusion, can you also provide results for the Weather dataset?
-	In my view, the authors must provide more detailed explanations for the proposed model to be self-contained. I think the current version is not easy to follow if readers were not already familiar with the Informer and the U-Net. In addition, the current version does not provide detailed information on dataset statistics (e.g. number of predictors and targets) and their pre-processing procedures.
-	Can you provide more in-depth experiment analyses for showing (1) how the U-Net shaped architecture helps long sequence time-series forecasting and (2) how does it affect the computational and memory efficiency of the model.


**Summary Of The Paper:**

Recent works such as the Informer have used efficient attention mechanisms and shown significant performance improvements in the long sequence time-series forecasting problems. However, the authors argued that using only the coarsest past representations for the decoder could be a major limitation. In this paper, the authors proposed the Yformer model by combining the Informer and the U-Net architectures. They adopted direct connections from the multi-resolution encoder to decoder to leverage both coarse and fine-grained representations. The authors claimed the effectiveness of the proposed method through three benchmark datasets used in the Informer paper.

**Summary Of The Review:**

While the proposed methods hold great promise, it has several issues to be addressed regarding the fairness of the experiments, a missing experiment dataset, and more detailed explanations to be self-contained.

---

> ### Author Response · Authors · 2021-11-16
> **Rebuttal**
>
> First of all, we would like to sincerely thank the reviewer for providing a thorough insightful review. Below is a point-to-point response.
>
> **Q1:** The Informer does not rely on the future predictors x', doesn't this make the comparison unfair?
>
> **A:**  We appreciate your valuable feedback on the potential of Yformer architecture however, we kindly disagree with the idea that Informer and Yformer use different input data. Within the Informer paper [1] (Page 5: Decoder section), the authors mention that "X_0 is a place holder for target sequence" which might confuse the reader to believe that the Informer does not use any future information. However, in the following paragraphs, the authors mention that "The X_0 contains target sequence’s timestamp, i.e. the context at the target week.". The context of the target week contains information about the future time points (like the week, holidays, etc), which we call "future predictors" as these are features that provide information for a particular future time step. We do not consider any power load features as future predictors. Hence, both the Yformer and the Informer use exactly the same information making the comparison fair.
>
> **Q2:** The authors stated that they used four benchmark datasets. Could you add results for the weather dataset?
>
> **A:** We apologize for the confusion. We considered the varied frequency of ETTh1 named ETTm1 as the fourth dataset. Changing the time series frequency to generate a time series with new characteristics is used in [1, 2] and we followed the same. We would be providing the results for the Weather forecast for consistency with the Informer baseline. Thanks for the comment.
>
> **Q3:** The authors must provide more detailed explanations for the proposed model and dataset to be self-contained
>
> **A:** We acknowledge that the current draft is missing the data statistics and pre-processing steps, and we would be adding these details to the appendix sections soon. Our experimentation protocol (datasets, pre-processing) was kept exactly similar to the Informer[1] baseline for a fair comparison. Except for the hyperparameters mentioned in Appendix E, all the model parameters like the convolution filter size, etc., were kept exactly the same as the Informer. We also mention the main ideas behind the ProbSparse mechanism and U-Net in the main text for completeness. We would greatly appreciate it if you could point us to the additional information required from the Informer or the U-Net paper.
>
> **Q4:**  How the U-Net-shaped architecture helps long sequence time-series forecasting?
>
> **A:** In Figure 2, we show how the U-Net architecture (Yformer) helps in reducing the error with respect to the baseline informer. The results show that especially for larger horizons the U-Net-based Yformer achieves lower error in comparison to the Informer baseline. Furthermore, as previous reviews pointed out, an ablation showing the Yformer model with and without skip connection would be presented soon.
>
> **Q4:**  How does U-Net affect the computational and memory efficiency of the model?
>
> **A:** We thank the reviewer for this great suggestion. Currently, we provide a short description of U-Net-based Yformer model complexity within the Methodology Section (Page 5, paragraph 2). However, we will provide a more comprehensive comparison with the Informer as a section within the appendix.
>
> We would like to thank you once again for taking the time to provide a thorough review.
>
> References
>
> [1] Zhou, H., Zhang, S., Peng, J., Zhang, S., Li, J., Xiong, H., & Zhang, W. (2021). Informer: Beyond Efficient Transformer for Long Sequence Time-Series Forecasting. Proceedings of the AAAI Conference on Artificial Intelligence, 35(12), 11106-11115
>
> [2] Salinas, David, et al. "DeepAR: Probabilistic forecasting with autoregressive recurrent networks." International Journal of Forecasting 36.3 (2020): 1181-1191.

---

> ### Comment · Reviewer_HmTV · 2021-12-05
> **Post-Rebuttal**
>
> Thank you for the detailed feedback on my comments. Although the authors have addressed my comments, unfortunately, I also agree with some of the concerns and comments from the other reviewers. Since the authors did not provide a rebuttal for the other reviewers, I am maintaining my original score.

---

### Official Review · Reviewer_yrSd · 2021-11-01

**Correctness:** 3
**Technical Novelty And Significance:** 2
**Empirical Novelty And Significance:** 3
**Recommendation:** 3
**Confidence:** 3

**Main Review:**

Strengths
---
Overall, the proposed architecture is intuitively compelling – echoing innovations observed in multi-horizon forecasting architectures (see related works comment below), while improving computational complexity using ProbSparse attention and downsampling. The strong improvements over the Informer baseline in numerous experiments also convincingly demonstrate the benefits of the proposed model for the LSTF problem.

Weaknesses
---
However, there are several key limitations that need to be addressed before the paper can be recommended for acceptance:
1.	**Architectural details** – While the network diagrams and descriptive text do a good job in providing a high-level overview, the lack of details make it difficult to evaluate the architecture in depth. For instance, a couple of questions come to mind:
  *	Do all historical input features need to be known in the future? The problem formulation is confusing here as x and x’ are R^{* x M} which seems to imply identical lengths T and number of features T – despite the text mentioning x’ is from T to T+tau.
  *	How are dimensions modified in each layer of the network? As the downsampling/upsampling parallels to U net appear to be a key part of the model, details on how this is performed is important.
  *	What are the keys, queries and values used for each attention layer (ProbSparseAttn, MaskedAttn, ProbSparseCrossAttn), and how is ProbSpraseCrossAttn implemented concretely?
  *	What is the length of the Conv1d filters in the various blocks, and are they purely linear transformations? Do dimensions change between each transformation?
  *	Is masked self-attention essential in the Y-Future encoder, and any reason why ProbSparse is not preferred? Does this affect computational efficiency, given that forecasting horizons appear to be larger than history lengths in many experiments from Appendix E.2?
2.	**Related works** -- While the authors do a good job of citing models for LSTF, the paper lacks references to modern neural forecasting architectures, many of which are attention-based and show improvements over LogTrans [2, 3] and DeepAR [1-3]. While computationally more inefficient, they also contain similar modifications to those proposed by the YFormer. For instance, [2, 3] use of distinct encoding mechanisms for historical inputs, future inputs, and static variables -- all of which are fed into a common attention-based decoder. In addition, [1] also trains the network using past targets as a regulariser (backcast). Comparisons to these models would help to further motivate the YFormer architecture as well.
3.	**Benchmarks** -- Given the focus on LSTF, comparison to simpler architectures that allow for extended receptive fields, e.g. dilated convolutions with WaveNet, would be useful. This is particularly important for time series datasets, which can be prone to overfitting with complex models -- as shown by the short-horizon outperformance of DeepAR on the ECL dataset in the Informer paper.

Typos
1.	DeepAR is also mentioned as a benchmark, although results are not included in the paper.

References
1.	Oreshkin et al. N-BEATS: NEURAL BASIS EXPANSION ANALYSIS FOR INTERPRETABLE TIME SERIES FORECASTING. ICLR 2020.
2.	Lim et al. TEMPORAL FUSION TRANSFORMERS FOR INTERPRETABLE MULTI-HORIZON TIME SERIES FORECASTING. International Journal of Forecasting, Volume 3 Issue 4, 2021.
3.	Eisenach et al. MQTRANSFORMER: MULTI-HORIZON FORECASTS WITH CONTEXT DEPENDENT AND FEEDBACK-AWARE ATTENTION. Arxiv 2020.


**Summary Of The Paper:**

The authors propose a new Transformer-based architecture for long-sequence temporal forecasting (LSTF) utilising ProbSparse attention mechanisms to efficiently capture long-term dependencies with L log(L) complexity.

The Yformer builds on the Informer architecture with 3 key innovations:
1.	Using distinct encoders to capture historical and known future information separately. This improves representation learning for time series data, while still maintaining computational efficiency with ProbSparse attention.
2.	Using a common decoder to process encoder representations jointly. The is also contains an upsampling step inspired by U-Net, although the benefits of upsampling are not explicitly evaluated.
3.	Including an auxiliary reconstruction loss which uses the reconstruction error of past targets to regularise training.


**Summary Of The Review:**

While the results do clearly show improvements in both forecasting performance and computational efficiency, many additional details on the architecture need to be included before the paper is ready for publication.

---

### Official Review · Reviewer_p7jA · 2021-11-02

**Correctness:** 4
**Technical Novelty And Significance:** 3
**Empirical Novelty And Significance:** 3
**Recommendation:** 5
**Confidence:** 3

**Main Review:**

Strength:
1. The paper is well written and the proposed framework is easy to understand. In addition, I believe the mathematical description of the model is correct.
2. Extensive evaluation is conducted, and the proposed YFormer shows an average of over 10% improvement compared with state-of-the-art models.
3. A good ablation study is provided to justify the choice of the proposed architecture, and also hyper parameter selection.
4. The authors of this paper choose baseline very carefully. They mentioned the reason why they are comparing with certain baseline models, identified some issues in some of the baseline models, and also provided reasoning why models such as Query Selector is not being used as a baseline model. I believe this thorough investigation and understanding of previous works is very important.

Weakness:
1. The mathematical description of the proposed architecture and task, although correct, is a bit over complicated. For example, section 3 describes a standard time-series forecasting problem with its corresponding notations. I would encourage the authors to review the notations needed in this section. I think some of them are not being used afterwards.
2.Since the results provided is an average of three runs. It would be beneficial if the authors could provide the standard deviation of the results as well. It would be informative to have an estimation of the variance of the proposed model.
3. In the abstract, the authors claim the model is tested on four datasets, in section 6.1,  it is said to be two real-world public datasets and one public benchmark. However, (I could be missing it somewhere), seems like only two datasets - ETT and ECL, are being evaluated on.

**Summary Of The Paper:**

A Former model is proposed in this paper, based on a Y-shaped encoder-decoder architecture that(1) uses direct connection from the downscaled encoder layer to the corresponding upsampled decoder layer in a U-Net inspired architecture, and (2) combines the downscaling/upsampling with sparse attention to capture long-range effects, and (3) stabilizes the encoder-decoder stars with the addition of an auxiliary reconstruction loss. The proposed model is evaluated on ETT and ECL dataset, and showed superior performance against baseline models including LogTransformer, LSTnet, Informer and Informer*.

**Summary Of The Review:**

Overall I think this is a good paper. Please refer to the above section for detailed review.

---

> ### Comment · Reviewer_p7jA · 2021-11-28
> **Revising recommendation score.**
>
> Since the authors did not provide a rebuttal to address some of the comments. I believe the paper would be stronger if the authors could pay more attention to the correctness of claims (including which dataset and which benchmark is used) and consistency of the paper. Therefore, I'm revising my recommendation to score 5.

---

### Official Review · Reviewer_yQSB · 2021-11-08

**Correctness:** 3
**Technical Novelty And Significance:** 2
**Empirical Novelty And Significance:** 2
**Recommendation:** 3
**Confidence:** 4

**Details Of Ethics Concerns:**

N.A.

**Main Review:**

Strengths
* Llong range time series forecasting is an interesting problem to investigate.
* Adding skip connections between encoder and decoder is technically sound
* The overall experiment results showed the effectiveness of the proposed method.

Weaknesses
* The organization of this paper is not well. Many technical details are not very clear in the main context
* The overall technical novelty is limited
* The effectiveness of the skip connections are not fully assessed
* More details of the experiments are not provided.

The main problem of this paper is that the main context (especially the methodology section) is not self-contained. The reader will have to rely on details in the appendix or other papers to fully understand the proposed technique.

Another concern is the novelty. Skip connections are common practice in U-net and the idea of stabilizing the encoder and decoder by reconstructing the recent past is also not new. Although it is a new application area for skip connections, the overall technical novelty is limited.

In addition, the ablation study over whether the Skip connections are used or not is not provided.

Several related works are not mentioned or compared.

[1] "Think globally, act locally: A deep neural network approach to high-dimensional time series forecasting." Sen, Rajat, Hsiang-Fu Yu, and Inderjit S. Dhillon NeurIPS 2019.

[2] "Modeling long-and short-term temporal patterns with deep neural networks." Lai, Guokun, Wei-Cheng Chang, Yiming Yang, and Hanxiao Liu, SIGIR 2018.

[3] "Shape and time distortion loss for training deep time series forecasting models." Vincent, L. E., and Nicolas Thome NeurIPS 2019.

As for the experiments:
1. It is not clear whether the setting in Eq. (1) is consistent with the settings in Informer or Reformer.
2. It is also not clear how to set y’ in the experiments.
3. Only two datasets are used for evaluation, which may not be sufficient to show the generalization capability of the proposed technique.
4. Standard deviations of the prediction results are not provided.


**Summary Of The Paper:**

This paper presents Yformer to perform long sequence time series forecasting. The key idea is to employ skip connection to improve the prediction resolution and stabilize the encoder and decoder by reconstructing the recent past. The experiment results on two datasets showed the effectiveness of the proposed method.

**Summary Of The Review:**

See above.

---

### Decision · Program_Chairs · 2022-01-20

**Decision:**

Reject

**Comment:**

This paper presents Yformer to perform long sequence time series forecasting based on a Y-shaped encoder-decoder architecture. Inspired by the U-Net architecture, the key idea of this paper is to improve the prediction resolution by employing skip connection and to stabilize the encoder and decoder by reconstructing the recent past. The experiment results on two datasets named ETT and ECL partially showed the effectiveness of the proposed method.

Reviewers have common concerns about the overall technical novelty, presentation quality, and experiment details. The authors only provided a rebuttal to one reviewer and most concerns from the other three reviewers were not addressed in the rebuttal and discussion phase. The final scores were unanimously below the acceptance bar.

AC read the paper and agreed that, while the paper has some merit such as an effective Yformer model for the particular problem setup, the reviewers' concerns are reasonable and need to be addressed in a more convincing way. The weaknesses are quite obvious and will be questioned again by the next set of reviewers, so the authors are required to substantially revise their work before resubmitting.